# Scrapie at Abattoir: Monitoring, Control, and Differential Diagnosis of Wasting Conditions during Meat Inspection

**DOI:** 10.3390/ani11113028

**Published:** 2021-10-21

**Authors:** Alexandra Esteves, Madalena Vieira-Pinto, Hélder Quintas, Leonor Orge, Adelina Gama, Anabela Alves, Fernanda Seixas, Isabel Pires, Maria de Lurdes Pinto, Ana Paula Mendonça, Carla Lima, Carla Neves Machado, João Carlos Silva, Paula Tavares, Filipe Silva, Estela Bastos, Jorge Pereira, Nuno Gonçalves-Anjo, Paulo Carvalho, Roberto Sargo, Ana Matos, Luís Figueira, Maria dos Anjos Pires

**Affiliations:** 1Animal and Veterinary Research Centre (CECAV), University of Trás-os-Montes and Alto Douro (UTAD), 5000-801 Vila Real, Portugal; mmvpinto@utad.pt (M.V.-P.); leonor.orge@iniav.pt (L.O.); agama@utad.pt (A.G.); aalves@utad.pt (A.A.); fseixas@utad.pt (F.S.); ipires@utad.pt (I.P.); lpinto@utad.pt (M.d.L.P.); fsilva@utad.pt (F.S.); jorgecpereira599@gmail.com (J.P.); roberto.sargo@gmail.com (R.S.); 2Centro de Investigação de Montanha (CIMO), Instituto Politécnico de Bragança, Campus de Santa Apolónia, 5300-253 Bragança, Portugal; helder5tas@ipb.pt; 3Pathology Laboratory, UEISPSA, National Institute for Agricultural and Veterinary Research (INIAV), I.P., 2780-157 Oeiras, Portugal; paula.mendonca@iniav.pt (A.P.M.); carla.neves@iniav.pt (C.N.M.); joao.silva@iniav.pt (J.C.S.); paulo.carvalho@iniav.pt (P.C.); 4Pathology Laboratory, UEISPSA, National Institute for Agricultural and Veterinary Research (INIAV), I.P., 4485-655 Vila do Conde, Portugal; carla.lima@iniav.pt (C.L.); paula.tavares@iniav.pt (P.T.); 5Centre for the Research and Technology of Agro-Environmental and Biological Sciences (CITAB), University of Trás-os-Montes and Alto Douro (UTAD), 5000-801 Vila Real, Portugal; ebastos@utad.pt; 6Genetic Department, University of Trás-os-Montes and Alto Douro (UTAD), 5000-801 Vila Real, Portugal; nuno_anjo@hotmail.com; 7Research Center for Natural Resources, Environment and Society (CERNAS), Polytechnic Institute of Castelo Branco (IPCB), 6000-767 Castelo Branco, Portugal; acmatos@ipcb.pt; 8Quality of Life in the Rural World (Q-Rural), Polytechnic Institute of Castelo Branco (IPCB), 6000-767 Castelo Branco, Portugal; lmftfigueira@gmail.com

**Keywords:** wasting disease, scrapie, abattoir, differential diagnosis

## Abstract

**Simple Summary:**

Observation of small ruminants showing differing degrees of leanness or emaciation is a common occurrence at slaughterhouses. Although either condition may result from similar causes, emaciation and cachexia are usually pathological conditions, warranting total condemnation of the carcass during post-mortem inspection. This is done in order to prevent unfit meat from entering the human food chain. Scrapie, a naturally occurring transmissible spongiform encephalopathy in small ruminants, is characterized by loss of body conditioning and a wasting appearance. Since atypical scrapie has been identified as occurring at a low, but very consistent, the prevalence in small ruminant populations, it is advisable to include this disease as a differential diagnosis of wasting conditions detected during meat inspection at the abattoir. Vigilance for detection of putative scrapie in slaughterhouses, and its differential diagnosis from other conditions associated with wasting carcasses, are of paramount importance for post-mortem decisions on the fate of carcasses, offal, as well as animal by-products.

**Abstract:**

Wasting disease in small ruminants is frequently detected at slaughterhouses. The wasting disorder is manifested by the deterioration of the nutritional and physiological state of the animal indicated by thinness, emaciation, and cachexia. Evidence of emaciation and cachexia, alone, are pathological conditions leading to carcass condemnation during an inspection. Several diseases are associated with a wasting condition, including scrapie, pseudotuberculosis, tuberculosis, paratuberculosis, Maedi Visna, and tumor diseases. On the other hand, parasitic diseases, nutrition disorders, exposure or ingestion of toxins, metabolic conditions, inadequate nutrition due to poor teeth, or poor alimentary diet are conditions contributing to poor body condition. Classical and atypical scrapie is naturally occurring transmissible spongiform encephalopathies in small ruminants. The etiological agent for each one is prions. However, each of these scrapie types is epidemiologically, pathologically, and biochemically different. Though atypical scrapie occurs at low incidence, it is consistently prevalent in the small ruminant population. Hence, it is advisable to include differential diagnosis of this disease, from other possibilities, as a cause of wasting conditions detected during meat inspection at the abattoir. This manuscript is a review of the measures in force at the abattoir for scrapie control, focusing on the differential diagnosis of gross lesions related to wasting conditions detected in small ruminants during meat inspection.

## 1. Introduction

Antemortem and post-mortem inspection of animals is an important process for detecting zoonotic diseases and is a vital component of the One Health concept. Moreover, antemortem and post-mortem inspection are also inherently important sentinel tools for the detection of animal diseases, in general, requiring additional diagnostic tests for suspicious cases.

Due to the relatively low economic value of small ruminants, post-mortem examination is an inexpensive and commonly used tool for monitoring diseases or other conditions. Post-mortem inspection is the most useful tool for achieving a pathological diagnosis and, in many cases, the only way to identify the majority of lesions on these animals. To avoid overlooking gross pathological findings, the post-mortem inspection must be performed methodically, always following the same approved and certified procedures. Visual inspection can be complemented with additional diagnostic tools. The European Food Safety Authority (EFSA) regarding human health hazards during inspection of meat from sheep and goats recommends omitting palpation and incisions to the greatest extent possible. This recommendation renders the post-mortem inspection as being essentially a visual procedure. However, the official veterinarian practice dictates additional post-mortem inspection procedures in certain instances. These additional procedures require incision and palpation of carcasses and offal when there is any indication of a possible risk to human or animal health and welfare [1]. Presentation of a wasting condition in a carcass should be considered one of those exceptions to relying on a visual diagnosis, alone. This is because the wasting condition may have a multiplicity of origins, including the result of an infectious agent or deficient animal welfare.

The wasting condition can be commonly observed in small ruminants during meat inspection. The wasting condition can result from poor nutrition, emaciation, or cachexia. Although emaciation and cachexia can result from similar etiologies, they are pathological conditions resulting in carcass condemnation during meat inspection. In animals designated as thin, there is a marked reduction of fat, but, otherwise, the animal has a normal appearance and texture [2]. Emaciation, compared to thinness (the two should not be confused) is a pathological condition involving atrophy of muscular tissue in addition to a reduction in the amount of fat tissue which acquires an abnormal oedematous and jelly-like consistency with an off-color, yellowish appearance (serous fatty atrophy) [2,3,4]. Cachexia is the most severe case of extreme emaciation. With this condition, there is also fatty tissue atrophy but additional atrophy of muscles and liver, and shrunken lymph nodes [2,4]. In the most severe cases of cachexia, there is serous atrophy of bone-marrow fat resulting in it appearing as a red, gelatinous mass [2,3].

Several conditions and diseases can be related to emaciation, including scrapie (classical and atypical). This disease presents as clinical signs of changes in typical animal behavior, chiefly incoordination of gait and trembling. Classical and atypical scrapie in small ruminants is naturally occurring transmissible spongiform encephalopathies (TSEs) caused by prions. The two TSEs are epidemiologically, pathologically, and biochemically different. Since atypical scrapie has a low, but consistent, the incidence in small ruminant populations, it is advisable to include this disease for differential diagnosis of wasting carcasses detected during post-mortem inspection. Generally, this condition is linked with chronic wasting diseases such as Maedi-Visna, paratuberculosis, caseous lymphadenitis, tuberculosis, and severe cases of parasitic diseases, tumor diseases, toxic or metabolic conditions, or inadequate nutrition due to poor teeth or poor alimentary diet.

This manuscript reviews clinical signs and indicator lesions to differentiate the causes of wasting conditions in carcasses of small ruminants at the slaughterhouse. The differential diagnosis in the slaughterhouse of gross lesions associated with the wasting condition is fundamental to food safety, including determination of treatment and destination of animal by-products. The main focus of this paper is to emphasize differential diagnosis of putative scrapie relative to other wasting conditions/diseases. The review also covers recommended certified laboratory tests and measures at the abattoir for scrapie control.

## 2. Scrapie

Classical scrapie (CS), the ancient animal transmissible spongiform encephalopathy affecting sheep and goats, is a naturally transmitted infectious and fatal neurodegenerative disease, reportable to the World Organization for Animal Health (OIE). The disease state results from the accumulation of a misfolded form (proteinaceous infectious particle, prion, or PrP^Sc^) of a normal host glycoprotein (cellular prion protein PrP^c^) encoded by the PrP gene (prnp). The first reported case of CS occurred in the United Kingdom in 1732. This disease was transmitted further through the export of live animals to western European countries, eventually reaching a worldwide distribution, except for Australia and New Zealand [5,6]. CS can result in serious economic loss due to decreased production, export loss, and increased cost for carcass disposal [5].

Under natural exposure, infection of CS mainly occurs during the period of birth (maternal-lateral transmission). This is followed by a long asymptomatic incubation period, usually ranging between two and seven years, during which infected animals are a source of further contamination. The clinical phase of the disease can last several months. Initially, the disease is characterized by a progressive loss of body weight (with preserved appetite) and slight behavioral changes, occurring for short periods that may go unnoticed. Later stages of the disease present progressive pruritus accompanied by hyperesthesia and uncoordinated gait [7].

In 1998, a different form of sheep prion disease (named Nor98 or atypical scrapie, AS) was identified in Norway [8,9]). Retrospective studies of AS, however, showed the disease had been present since, at least, 1972 [10]. The two forms of scrapie differ in clinical presentation, lesion profile, and distribution of PrP^Sc^ within affected sheep, as well as genotype of animal affected and associated epidemiology (Table 1). Atypical scrapie has been identified worldwide, including New Zealand [11] and Australia [12]. Global distribution of AS exhibiting similar incidence and epidemiology within a given flock suggests AS has a spontaneous etiology or is a poorly transmitted TSE under natural conditions. Thus, in contrast to CS, AS is considered a non-reportable disease by the World Organization for Animal Health (OIE) [5,7,13].

Epidemiological studies have failed to confirm exposure to small ruminants, or their products is a risk factor for zoonotic spill-over of prionic diseases to humans. This finding has led to the conclusion small-ruminant prion-diseases present a negligible risk of developing into a zoonotic outbreak. However, recent findings show the prevalence and geographical distribution of TSEs in small ruminants have been underestimated, in addition to the observation of transmission of CS in non-human primates [14] and human PrP transgenic mice [15]. Such new findings warrant an urgency to reassess the zoonotic potential of CS and AS under conditions of field exposure [6]. Furthermore, recent studies showed transmission of the classical-BSE (c-BSE) prion to bovine transgenic mice after inoculation with AS isolates. This finding shows c-BSE might be present in natural AS isolates as a secondary infectious agent [16]. Such recent observations raise serious concerns for maintaining TSE surveillance in ruminants. Moreover, it argues for implementing effective measures (Specific Risk Material removal and feed ban on animal proteins) established, from the time the BSE crisis first appeared, in order to prevent human exposure to these pathogens. Additionally, these measures will prevent further exposure of farm animals to prions, including AS, potentially circulating in farm animal populations [6,16].

Scrapie can be caused by several strains of prions, but the diversity of these strains in small ruminants remains undefined. At least five different ovine prionic strains are known, including AS [16]. These strains may exhibit different disease phenotypes and pathogenesis. As already mentioned for CS, TSEs, in general, in small ruminants are characterized by long asymptomatic incubation periods usually ranging between two and seven years, during which time infected animals are a potential source of contamination [21] (Table 1).

All Member States (MS) of the European Union (EU) developed and implemented scrapie surveillance, control, and eradication programs, respective to each EU country. These include passive surveillance, involving the investigation of animals with neurological signs, and active surveillance, entailing sampling of small ruminant populations from specific groups of animals based on fallen stock, those for emergency slaughter, animals with clinical symptoms of diseases other than TSEs during the antemortem inspection, and healthy animals slaughtered for human consumption (Figure 1). The eradication program focuses on small ruminants identified as at-risk cohabitants of confirmed classical scrapie cases. This program is under constant review as new scientific knowledge on the pathology and epidemiology of these diseases come to light [22,23]. Currently, there is no validated laboratory test for the diagnosis of TSEs for living livestock. Definitive diagnosis of scrapie is based on post-mortem laboratory tests of samples from the central nervous system (CNS).

CS is still the most frequently reported type of prionic disease in the EU in goats and sheep, but with a differing incidence between member states. Over the past 10 years (2010 through 2019), the incidence per 10,000 tested animals, considering both CS and AS and both goats and sheep, showed a range between 0.5 and 5.4. However, there has been a statistically significant decreasing trend for both CS and AS in sheep, but not yet verified in goats [24]. International policies favor higher commercial value on free-roaming flocks of small ruminants. As such, it is important to ensure these flocks are certified to be free of CS, where animal movement is a higher risk for transmission. Thus, COMMISSION REGULATION (EU) No 630/2013 [25], amended Regulation (EC) No. 999/2001 [22], stipulates to only maintain live animals for breeding purposes, for intra-Community exchange and export, from holdings classified as according to its risk status concerning CS. A classification scheme for small ruminants has been defined according to CS in holdings of negligible risk, controlled risk, and undetermined risk, including restricted movement and continued surveillance. Animal identification involving traceable records of entry and exit of sheep and goats from farms, testing for CS in animals over 18 months of age, as well as preference for ARR/ARR resistant sheep, are fundamental conditions of this revised regulation.

An essential component for the control of TSEs in slaughterhouses also includes correct classification and consequent processing of animal by-products. Concerning the zoonotic potential of classical BSE, the EFSA amended prior requirements for the removal of specified risk material (SRM) in small ruminants in the slaughterhouse. The amended procedure recommends those tissues having the highest levels of BSE infection in small ruminants be designated as SRM according to their classification in Category 1 by-products [26]. In this regard, only the skull, including brain and eyes, and the spinal cord of slaughtered animals over 12 months of age, as confirmed by a competent Member State authority, or showing a permanent incisor penetrating the gum, should be considered as SRM in ovine and caprine animals.

According to Regulation (EC) No 1069/2009 [27], the entire carcass and all body parts, including hides and skins, of animals suspected of being infected by a TSE are to be discarded. In these cases, the presence of a TSE must be officially diagnosed by laboratory confirmation, or the animal has been killed in the context of TSE eradication measures. In such cases, these are categorized as Category 1 by-products to be destroyed by incineration.

## 3. Scrapie—Differential Diagnosis and Judgment during Meat Inspection

The antemortem clinical presentation of the two types of scrapie can include weight loss or emaciation, behavioral changes (fixed stare, isolation, hyperexcitability, loss of inquisitiveness), trembling, and uncoordinated gait. Pruritus may also be present in CS, as opposed to AS. As these clinical signs may be considered as suggestive for scrapie, at the abattoir, varying combinations of these signs and degree of severity can be observed. In some circumstances, a combination of these clinical signs may not be scrapie-specific, leading to misdiagnosis. For example, staggering, self-isolation, and trembling are signs that can also be present in other wasting diseases, most notably in a weak condition. In these cases, other antemortem signs (e.g., joint swelling, lameness, respiratory or digestive signs) and post-mortem findings (e.g., abscesses, granulomatous lesions, nodular lesions in different locations) could be of significant importance for directing a diagnosis, in view of emaciation in the absence of other lesions, on post-mortem inspection, is an indication of scrapie. Antemortem inspection is highly important for detecting and registering any neurological disease that may not present any observable macroscopic causation during post-mortem inspection. Such cases could be scrapie. But, in some circumstances, when clinical signs are not obvious or antemortem inspection (e.g., overloaded lairage) does not permit detection of neurological abnormalities, suspected cases of scrapie must be based on detection of wasting carcasses.

Post-mortem inspection may only present emaciation of carcasses as the only finding of either form of scrapie. But, as aforementioned, wasting conditions can result from a multiplicity of factors, namely infectious diseases, or deficient animal care (management/nutritional deficiency). Thus, pathological observations on post-mortem inspection may be a decisive factor for differential diagnosis.

A graphical categorization of the main wasting diseases/conditions, other than scrapie, seen during meat inspection is presented in Figure 2. Clinical signs (antemortem) and gross pathological findings (post-mortem) of these non-scrapie wasting conditions are described further, in the following sections (Section 3.1, Section 3.2 and Section 3.3).

Selection of diseases/conditions associated with wasting conditions was established through thorough discussions with a group of multidisciplinary stakeholders (academics, meat inspection researchers, official meat inspectors, practitioners specialized in small ruminant diseases, pathologists, veterinary public health specialists, and geneticists), based on the totality of condemnation data at the abattoir and respective anatomopathological and microbiological findings, in Portugal.

### 3.1. Other Wasting Diseases

Several infectious diseases can lead to a chronic wasting syndrome, characterized by progressive weight loss, including the following: caseous lymphadenitis (CLA), pseudotuberculosis [28,29,30], tuberculosis [31,32], and paratuberculosis [33] or even viral diseases Maedi Visna [34].

The main antemortem signs and gross post-mortem lesions observed in animals/carcasses with corresponding chronic wasting from infectious (bacterial/viral) diseases are listed in Table 2.

The presence of malignant tumors in sheep and goats generally is associated with the declining condition. At the time they reach the slaughterhouse, animals exhibiting such tumors are generally relatively old (e.g., dairy production animals), having aged long enough to develop gross tumoural lesions [57]. There are reports of such tumors metastasizing and spreading to all organs and systems of sheep and goats [57,58,59,60], being more frequently found in the alimentary, lymphohematopoietic, respiratory, skeletal systems and liver [58]. The anatomical site of the tumor may influence the presentation of clinical signs observed during the antemortem inspection, including emaciation, or dyspnea in the case of lung tumors. In animals presenting tumors, the type and appearance of resultant lesions vary widely, depending on the affected system or organ, and occur as polyps, nodules, cystic, ulcers, tumor-masses, as single or multiple lesions, either limited to one organ or disseminated, as with distant metastasis. Certain lesions observed upon post-mortem inspection may be mistakenly diagnosed as scrapie. The presence of tumor-like lesions require the official veterinarian to evaluate if they are malignant or benign in order to apply the correct condemnation judgment: total for malignant and partial for benign. In all cases accompanied by emaciation, total condemnation is compulsory.

The majority of observed malignant tumors are spontaneous and their etiology has not been recognized so far, with the exception of two neoplasms, nasal carcinomas of sheep and goats, and ovine pulmonary adenocarcinoma (OPA). Pulmonary adenocarcinoma is commonly recorded as an incidental finding in the abattoir [61,62]. Sheep affected by OPA show an afebrile respiratory illness associated with loss of weight [63].

Clinical signs and gross lesions observed in emaciated small ruminants resulting from the presence of malignant tumors are outlined in Table 3.

### 3.2. Parasitic diseases

Subclinical and chronic parasitic infections are considered major causes of economic loss in small ruminants exhibiting poor physical condition and chronic wasting at pasture [64,67]. Symptoms of wasting and emaciation associated with various parasitic diseases are outlined in Table 4, including etiological agents, clinical signs, and gross lesions.

### 3.3. Nutritional Disorders

Nutritional disorders can also be associated with wasting carcasses of small ruminants [78,79]. In meat inspection, differential diagnosis with scrapie should be done in an antemortem examination since these situations are not accompanied by other post-mortem injuries in addition to carcass emaciation.

Primary nutritional conditions (disorders) leading to poor body condition may result from inadequate energy and protein levels or a lack of specific macro and microminerals. These inadequacies may be due to poisonous pasture plants, leading to direct toxicity of the animal, or deficiencies in soil nutrients for, or diseases in, forage plants, leading to repercussions in maintaining normal ruminal flora [60,78]. Crude protein levels below 7% in the diet of small ruminants negatively affect normal rumen bacterial activity, resulting in slowed growth, decreased immune function, anemia, edema, and death. Similarly, deficiency of sulfur in the diet may lead to anorexia, reduced weight gain, decreased wool growth, and possibly death. Copper deficiency may also result in waste. Deficiency of this mineral during early-stage growth shows characteristic clinical signs of incoordination in newborns—enzootic ataxia or swayback. Copper deficiency also increases susceptibility to gastrointestinal diseases contributing to emaciation, hypoproteinemia, and anemia [60,79,80]. Cobalt deficiency in sheep or goats is characterized by vitamin B12 deficiency, resulting from ruminal dysbiosis, accompanied by symptoms of hyporexia, emaciation, anemia, rough hair coat, and wasting disease. Cobalt deficiency is also associated with white liver disease, which may also include deficiencies in phosphorus and copper or chronic parasitism [81,82,83]. Hypovitaminosis A is an emaciation disease and is a condition associated with the absence of or limited access to fresh green forage. Clinical signs include convulsive seizures, weight loss, depressed immune function, decreased fertility, hair loss, night blindness, and ocular discharge [60,79].

Diagnosis of wasting as a result of nutritional deficiencies is not easily differentiated from scrapie at antemortem examination even if neurological abnormalities associated with scrapie are observed. Moreover, at post-mortem inspection of nutritional diseases, as with scrapie, other injuries or anomalies may not be easily detected. Neurological signs of scrapie in infected animals may not always occur, particularly in atypical scrapie. In such cases, laboratory diagnosis is essential.

Secondary nutritional disorders may be related to an inability to obtain or absorb nutrients from available food. Such deficiency may lead to a variety of problems, including locomotor problems (e.g., arthritis, traumatic osseous or nerve lesions, rickets, foot rot, foot scald, foot abscesses), blindness (e.g., polioencephalomalacia, hypovitaminosis A, infectious keratoconjunctivitis), diseases of the oral cavity (e.g., periodontal disease, gingivitis, tooth root abscesses, diseases associated to oral erosion or ulcers) and oral cavity anomalies (e.g., loss of teeth, traumatic lesions in the mouth, fractures of the maxilla and mandible; brachygnathism, prognathism, congenital absence of lower incisors or premolar, persistent deciduous incisors, dentigerous cysts). With secondary nutritional disorders, observation of antemortem signs is often associated with post-mortem lesions that serve as the basis for differential diagnosis of these nutritional disorders from scrapie.

### 3.4. Laboratory Diagnosis of Scrapie

Conditions for differential diagnosis of wasting condition at the slaughterhouse, based on antemortem signs and post-mortem gross lesions, are summarized in Figure 3, Figure 4 and Figure 5.

There is no validated laboratory test for the diagnosis of scrapie in living animals. Definitive diagnosis of scrapie currently requires post-mortem laboratory tests using tissue samples of the central nervous system (CNS) in authorized laboratories and/or in the national reference laboratory for animal TSEs. These tests are designed to detect the presence of PrP^Sc^ in these tissues, using an enzyme-linked immunosorbent assay (ELISA), immunohistochemistry, or western immunoblots. Histopathological methods may lead to detection of vacuolar lesions in the CNS [13] having particular importance for differentiating scrapie from other neurological disorders (e.g., listeriosis, Maedi-Visna, polioencephalomalacia) or extraneural diseases causing progressive loss of condition, weakness or skin lesions (paratuberculosis, endoparasites, ectoparasites).

There are a number of commercially available rapid test kits for detection of PrP^Sc^ approved by the European Commission, listed in Annex X of Regulation (EC) No. 999/2001 [84] as last amended [85].

## 4. Differential Judgment of Wasting Carcasses during Meat Inspection

The judgment (sanitary decision) if meat is fit for human consumption, has to be based on two fundamental aspects: (1) degree of deterioration of the carcass, and (2) cause of loss of body condition. A correct diagnosis of the causes of the wasting condition (differential diagnosis) is essential, since it has major consequences to the sanitary processing of the carcass, as implemented by the official veterinarian, as well as in the classification of the carcass for determining the subsequent destination of animal by-products.

Judgment is based on the degree of loss of condition, presence of concurrent diseases, and eventually results of laboratory examinations. Particular attention has to be given to extent of emaciation, presence/absence of edema, and current diseases. This condition, especially if not associated with concurrent diseases, is among the most difficult to assess during meat inspection.

This is particularly the case depending on the locality where official conditional approval for manufactured purposes and/or heat treatment is not authorized. Note that EU Meat Inspection Regulation (Commission Implementing Regulation 2019/627 (article 30, n° 2), only predicts conditional approval to “meat not generally infected with cysticercus in which parts uninfected may be declared fit for human consumption after having undergone a cold treatment”. This is particularly the case in regions where conditional approval for manufactured purposes and/or heat treatment is not authorized. Note that EU Meat Inspection Regulation (article 30, n° 2) (Commission Implementing Regulation 2019/627 [1], only predicts conditional approval in “meat with not generally infected with cysticercus on which the parts not infected may be declared fit for human consumption after having undergone a cold treatment”.

Codex Alimentarius Commission [86] recommended that “General chronic conditions such as anemia, cachexia, emaciation, loathsome appearance, degeneration of organs” warrant total condemnation. Also, according to Commission Implementing Regulation 2019/627 [1], (article 45, point p) fresh meat derived from emaciated animals is considered unfit for human consumption.

Total condemnation is always warranted if the wasting condition is caused by chronic infection, and laboratory examination established the presence of infection, recent use of antimicrobial substances, or drug residues. In borderline cases, when no concurrent diseases are identified, it is advisable to retain the carcass for 12 h. If after this period there is considerable drying of the body cavity with an absence of serous infiltration of muscles, and negative laboratory tests, the carcass may receive a more favorable judgment. In the absence of disease, conditional approval may be given with heat treatment [87].

Situations involving simple thinness, where there are no signs of carcass compromise (carcass infection), even after a concurrent disease and post-mortem inspection reveal localized lesions of one organ or part of the carcass, only the affected organ or portion of the carcass, and associated lymph nodes, shall be declared unfit for human consumption. Thinness may be physiologically normal [87] as occurring in senile animals. In such cases, if upon post-mortem examination there is no accompaniment by other types of injuries, the carcass may be approved for consumption as long as it is not classified as a “cachectic by senility” carcass. If the referred carcass is suspected or officially confirmed to be infected by a TSE disease (e.g., classical or atypical scrapie), the entire carcass and all body parts, including hides and skins, are condemned [27].

Animal products from animals not fit for consumption could be a potential source of risk to public and livestock health. Regulation (EC) N° 1069/2009 [27] introduced the classification of animal by-products into three categories that reflect the degree of risk posed to public and animal health, on the basis of risk assessments. Disposal and/or use of animal by-products are also dependent upon their categorization.

Table 5 presents the different categorizations of condemned wasting carcasses, affected organs, or part of the carcass according to Regulation (EC) No 1069/2009 [27].

## 5. Conclusions

The considerable seriousness in the detection of scrapie to public and livestock health, and its potential for severe economic loss, demands the participation of an Official Veterinarian at the abattoir in order to guarantee careful and detailed inspection procedures and issuance of sanitary decisions. The judgment made for the fate of carcasses and any by-products requires an assessment of information compiled during ante- and post-mortem inspections, under the purview of veterinarian expertise and analytical reasoning.

Many different situations may account for wasting detected in animal/carcasses at the abattoir. Some of these conditions are devoid of public/animal health concerns. However, among these diverse situations leading to a poor carcass condition, some may involve animal welfare (poor care), infectious diseases, or zoonoses, such as TSEs. Moreover, in some of these cases, a definitive diagnosis is not available by macroscopical pathological examination. In such cases, laboratory procedures may be required. The laboratory diagnosis of a scrapie suspicion is mandatory, as it does not cause macroscopic lesions to be observed on post-mortem inspection. Failure to use laboratory tests to diagnose scrapie would result in the misevaluation of the carcass, and consequent incorrect classification of by-products. Missing the diagnosis of scrapie can have serious repercussions in terms of animal health and zoonotic spill-over to humans.

## Figures and Tables

**Figure 1 animals-11-03028-f001:**
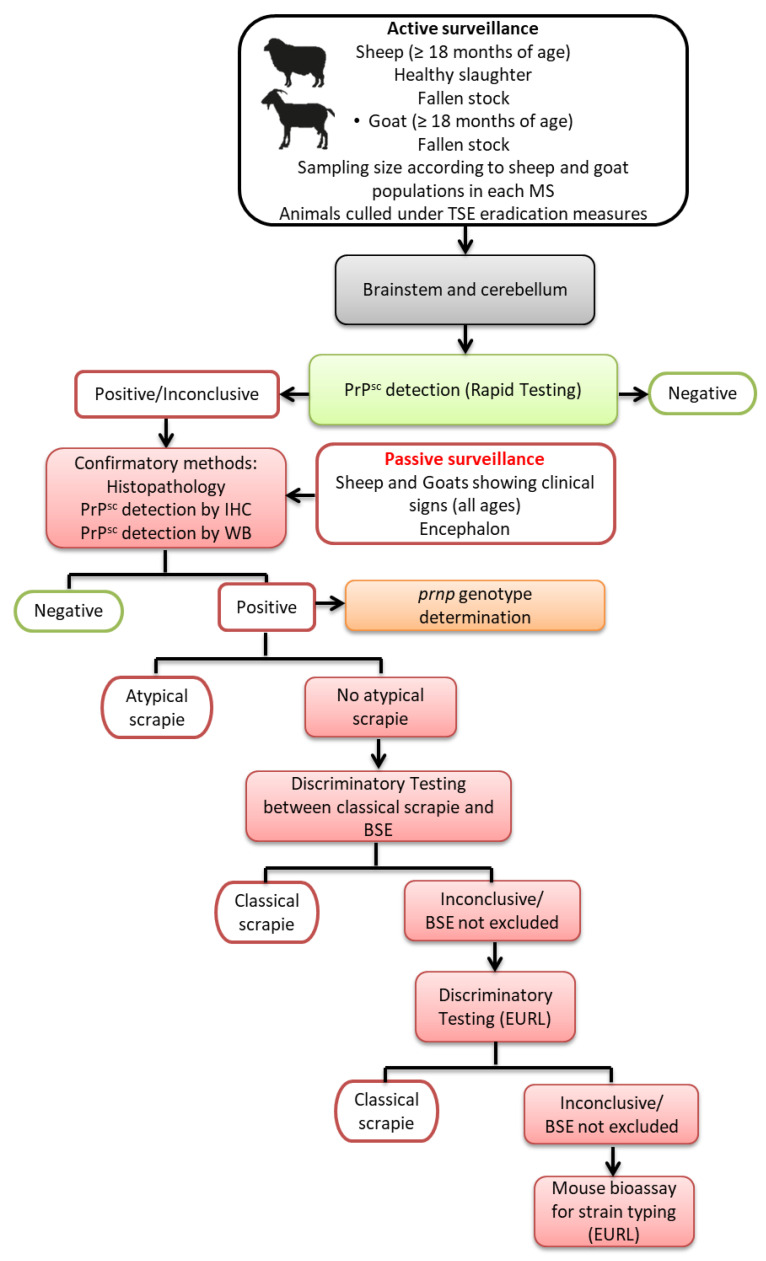
Sampling and diagnostic procedures in European Union TSE surveillance for small ruminants (MS—member state; IHC—immunohistochemistry; WB—Western Immunoblot; EURL—European Reference Laboratory for animal TSEs).

**Figure 2 animals-11-03028-f002:**
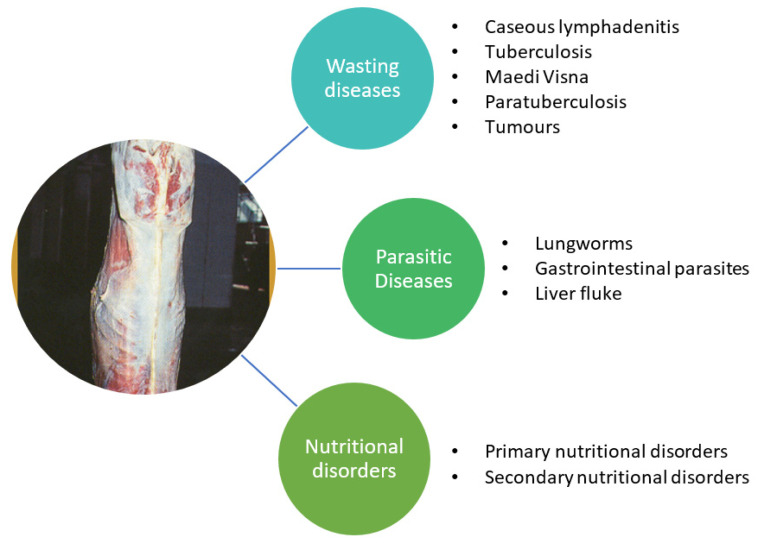
Categorization of the main non-scrapie diseases or conditions associated with a chronic wasting syndrome in small ruminants.

**Figure 3 animals-11-03028-f003:**
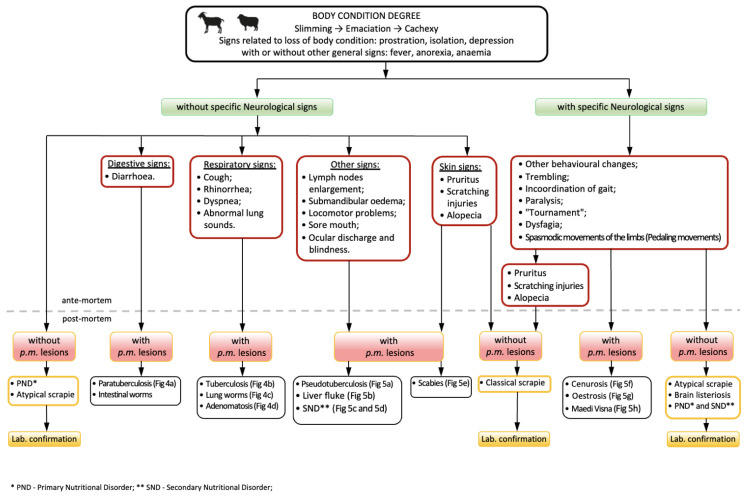
Sanitary Inspection of Wasting condition in small ruminants.

**Figure 4 animals-11-03028-f004:**
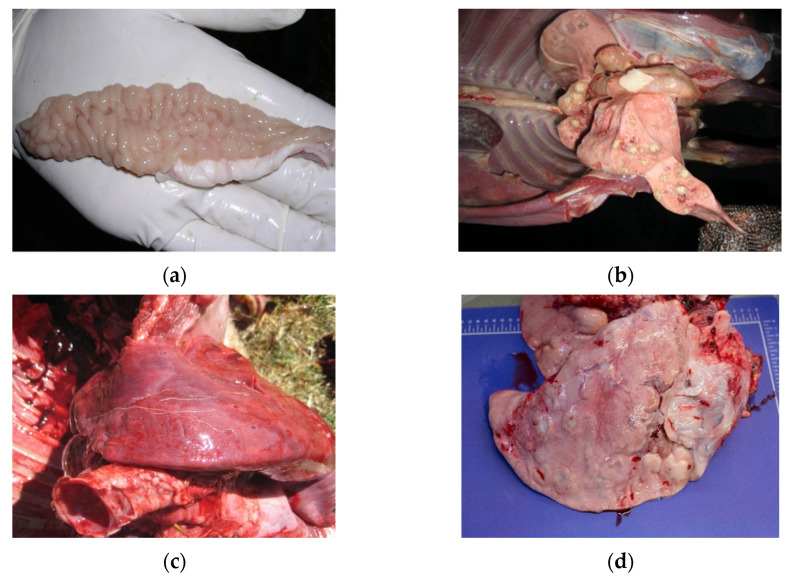
(**a**) Paratuberculosis; (**b**) Tuberculosis; (**c**) Lungworms; (**d**) Adenomatosis.

**Figure 5 animals-11-03028-f005:**
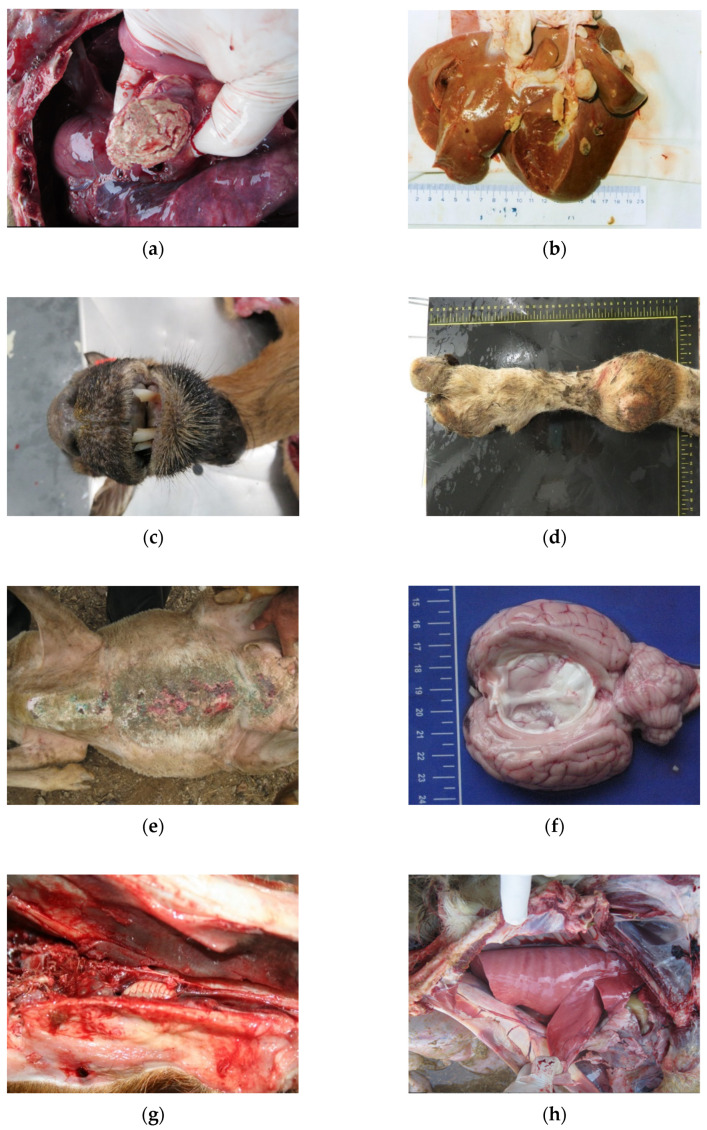
(**a**) Pseudotuberculosis; (**b**) Liver fluke; (**c**) Sore mouth; (**d**) Locomotor problems; (**e**) Scabies; (**f**) Cenurosis; (**g**) Oestrosis; (**h**) Maedi-Visna.

**Table 1 animals-11-03028-t001:** Key differential features between classical and atypical scrapie.

	Classical Scrapie	Atypical Scrapie
Epidemiology	Small ruminants between 2–5 years of age Several affected animals per flock Animal movement is a risk factor for the transmission	Small ruminants over 5 years of age Only one or two affected animals per flock Animal movement is not a risk factor
Role of *prnp* 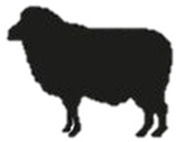	ARR/ARR Fully resistant ARR/AHQ, ARR/ARH, ARR/ARQ—Partially resistant AHQ/AHQ, AHQ/ARH, AHQ/ARQ, ARH/ARH, ARH/ARQ, ARQ/ARQ—Average risk of infectionARR/VRQ—High risk of infection AHQ/VRQ, ARH, VRQ, ARQ/VRQ, VRQ/VRQ—Highest risk of infection	Several genotypes including those related to resistance to classical scrapie AHQ/XXX, AFRQ/XXX—Highest risk of infection
Role of *prnp* 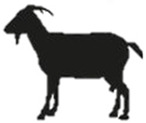	H154, Q211, K222—Potentially resistant NN146—Highest risk of infection	H154-Highest risk of infection
Clinical signs	Behavioral changes (fixed stare, isolation, hyperexcitability, loss of inquisitiveness) Trembling Uncoordinated gait **Weight loss or emaciation** Pruritus (main symptom in sheep, usually leads to wool loss) Positive scratch response (“nibble reflex”)	Behavioral changes (fixed stare, isolation, hyperexcitability, loss of inquisitiveness) Trembling Incoordination of gait **Weight loss or emaciation** Visual impairment
Spongiform distribution	Medulla oblongata (obex)	Cortices of the cerebellum and the cerebrum
PrP^sc^ types and distribution	PrP^sc^—Three-band profile on western blot between 27 and 19 kDa PrP^sc^ types: Intracellular: intraneuronal Extracellular: stellate, perivascular, subpial, fine granular, aggregates, plaque-like, linear, perineuronal, vascular plaques, subependymal, ependymal Medulla oblongata (obex) (+++) Placenta and peripheral and lymphoid tissues (except ARR/xxx)	PrP^sc^—Multiple-band profile on western blot between 31 and 11 kDa PrP^sc^ types: Extracellular: fine granular, aggregates, plaque-like, linear, perineuronal, Other: globular (white matter); punctate Medulla oblongata (obex) (+); Cerebellum (+++) No PrP^sc^ in peripheral and lymphoid tissues (but infectivity demonstrated)

A—alanine; R—arginine; H—histidine; Q—glutamine; V-valine; F—phenylalanine (codon 141); N—asparagine; clinical signs in bold are the most frequent; (+) mild; (+++) intense; [5,13,17,18,19,20].

**Table 2 animals-11-03028-t002:** Clinical antemortem signs and gross lesions were observed post-mortem on small ruminants with chronic wasting caused by infectious diseases.

**Caseous Lymphadenitis (CLA) or Pseudotuberculosis**		**References**
Etiology	*Corynebacterium pseudotuberculosis*	[28,29,30,35,36,37,38,39]
Clinical signs (Antemortem)	Swelling of affected superficial lymph node regions (e.g., retropharyngeal, submandibular, precrural, prescapular), emaciation and general deterioration of the animal.	
Gross lesions (Post-mortem)	Abscesses developing into pyogranulomatous lesions observed in one or more superficial lymph nodes (retropharyngeal, parotid, submandibular, popliteal, precrural, prescapular, or mammary). The spread of infection may extend to mediastinal and bronchial lymph nodes and multiple organs, such as the spleen, kidneys, and liver or mammary glands. Lesions may present as abscesses with a range of a few millimeters to centimeters in diameter, consisting of a creamy to thick, white-yellowish to pale-green purulent or caseated material surrounded by a thick fibrous wall; in mainly chronic cases, abscesses increase in size and develop a lamellated appearance (“onion ring”).	
**Tuberculosis**		**References**
Etiology	*Mycobacterium caprae*, *M. bovis*, occasionally other mycobacteria from *M. tuberculosis* Complex	[31,32,40,41,42,43,44,45,46,47,48,49]
Clinical signs (Antemortem)	Depending on the affected organs. Respiratory symptoms in the final phase of the disease: deep and productive chronic cough, with tachypnea, dyspnea, and abnormal lung sounds. Emaciation and general deterioration of the animal.	
Gross lesions (Post-mortem)	Pulmonary lesions (more frequent): range from pale nodules, with caseification necrosis, mainly in primary complex lesions. Miliary nodules, extensive granulomas, areas of coalescent caseification with softening of the caseous are usually associated with post-primary lesions. Liquefaction leads to the destruction of bronchial walls and cave formation. Lymph nodes can present radial caseification, with necrotic and calcified nodules and fibrosis. Generalized process: hepatic miliary lesions or large nodules, with a fibrous capsule; in serous membranes, the most frequent lesion is granulomatous serositis, the so-called “pearl tuberculosis”. Small granulomas of different sizes are observed, with caseification and calcification; lymphoid tissue and Peyer’s plaques can present thicker and firmer plaques, miliary foci, or caseous ulcers. Abomasum, rumen, reticulum, and omasum rarely present lesions.	
**Paratuberculosis**		**References**
Etiology	*Mycobacterium avium* subsp. *paratuberculosis*, allocated to two major strain types; type S (Sheep type with subtypes I and III) and type C (Cattle type or Type II; including type B: the USA and Indian Bison Type).	[33,50,51,52,53,54]
Clinical signs (Antemortem)	Initial signs can be subtle and become gradually more severe, leading to malnutrition, debilitation, and eventually death. Weight loss and exercise intolerance are prominent signs in sheep and goats. Diarrhea is less common compared with cattle, it is more likely to appear intermittently as soft feces. Some have submandibular edema, without other evidence of edema, with wool often damaged and easily shed. Anemia is also reported to be common.	
Gross lesions (Post-mortem)	Usually absent or subtle in subclinical carriers, and occasionally absent or minimal in clinical cases. Carcasses of animals with advanced disease are usually thin or emaciated, with loss of adipose tissue with serous atrophy of bone marrow (consistent with cachexia). Mesenteric, ileocolic, and colonic lymph nodes are moderate to severely enlarged and show a bulging cut surface, without evidence of necrosis or calcification. The intestine (distal part of the small intestine) exhibits mild to moderate diffuse thickening of the mucosa. There may be dependent edema and/or fluid in the body cavities.	
**Maedi-Visna**		**References**
Etiology	Retrovirus from the genus *Lentivirus*, an RNA virus, present five groups of strains, that included about 21 subtypes.	[33,55,56]
Clinical signs (Antemortem)	Slow progressive disease with indications of pneumonia and mastitis are the predominant clinical signs. Respiratory distress, non-painful hard udder (decrease of milk production). Clinical signs of arthritis are progressive with lameness. Neurological signs include ataxia, paresis, incoordination, and, in most severe cases, total paralysis, as a consequence of meningoencephalitis. Emaciation and general deterioration of the animal.	
Gross lesions (Post-mortem)	Lungs presenting diffuse pneumonia, with zones firm, discolored and enlarged, with grey spots on the pleural surface. The mediastinal lymph nodes are often enlarged. Non-suppurative interstitial mastitis. Central nervous system with chronic inflammatory lesions. Meningoencephalitis. Arthritis: The most affected joints are the carpal and tarsal, even metatarsal and metacarpal joints. Vertebrae may also be affected. Several multi-organ infections may be affected as the disease progresses. These include liver, kidney, and heart, with lymphoid tissue hyperplasia.	

**Table 3 animals-11-03028-t003:** Clinical signs and gross lesions associated with nasal carcinoma and pulmonary adenocarcinoma were observed in emaciated carcasses from small ruminants.

**Nasal Carcinomas of Sheep and Goats**		**References**
Etiology	Betaretroviruses in sheep (ENTV-1) and goats (ENTV-2)	[64]
Clinical signs (Antemortem)	Variable but include stertor, inspiratory dyspnoea, open-mouth breathing, nasal discharge, nasal deformity, and weight loss. Tumors arise from surface epithelium and glands of the ethmoidal conchae, entire nasal cavity possibly occluded when unilateral there is the deviation of the nasal septum with tumor protruding from the nostril.	
Gross lesions (Post-mortem)	Tumour may be unilateral or bilateral occupying the entire affected nasal cavity. The neoplastic tissue is white, firm, and multinodular, or may contain brown-red areas of hemorrhage and necrosis.	
**Ovine Pulmonary Adenocarcinoma (OPA)**		**References**
Etiology	Betaretrovirus, jaagsiekte sheep retrovirus (JSRV)	[63,65,66]
Clinical signs (Antemortem)	Afebrile respiratory illness, dyspnea, and moist respiratory sounds due to the accumulation of fluid in the respiratory airways and progressive weight loss.	
Gross lesions (Post-mortem)	Presence of grey or light-purple, firm coalescing nodules on lung, heavier lung. The cut surface of the lesion displays a granular appearance, is moist, and often exudes white frothy fluid as observed in respiratory passages, including the trachea. Intra and extrathoracic metastasis to lymph nodes and other tissues was also observed. In some countries, another form of OPA has been described (atypical OPA), in which non-coalescing, hard-white, and dry nodular lesions are a prominent feature.	

**Table 4 animals-11-03028-t004:** Clinical signs and gross lesions were observed post-mortem on emaciated carcasses from parasitized small ruminants.

**Lungworm Infection**		**References**
Etiology	*Dictyocaulus filaria*, *Muellerius capillaris*, *Protostrongylus rufescens*	[64,68,69,70]
Clinical signs (Antemortem)	Fever, cough, nasal discharge, moderate dyspnea, diarrhea, anorexia, tachypnea, weight loss, poor body condition, and chronic emaciation.	
Gross lesions (Post-mortem)	Presence of worms in bronchi or bronchiolar lumen, areas of atelectasis or consolidation secondary to bronchiolar obstruction, subpleural and pulmonary nodules (granulomas), and periodically in regional lymph nodes.	
**Gastrointestinal Parasites**		**References**
Etiology	Abomasal parasites: *Haemonchus contortus*, *Ostertagia (Teladorsagia) circumcincta* Intestinal parasites: *Trichostrongylus colubriformis*, *T. vitrinus*, *T. rugatus; Nematodirus spathiger*, *N. filicollis*, *N. abnormalis*, *Cooperia curticei*, *Bunostomum trigonocephalum*, *Oesophagostomum columbianum*, *O. venulosum*, *Eimeri ovinoidalis*, *E. weybridgensis/crandallis*, *E. ninakohlyakimovae*, *E. arloingi*	[44,64,67,71,72,73,74]
Clinical signs (Antemortem)	Anemia, hypoproteinemia accompanied by edema most prominently in the intermandibular space (bottle jaw), diarrhea, dehydration, poor body condition, and chronic emaciation.	
Gross lesions (Post-mortem)	Pale carcass, generalized edema, hydrothorax, hydropericardium, and ascites, liver pale and friable. Abomasal parasites: Foci of mucosal hemorrhage, presence of nematodes, content fluid, and dark red-brown owing to the presence of blood (*Haemonchus contortus*); multinodular raised pale areas, often with a depressed center on thickened mucosa (*Ostertagia*). Intestinal parasites: villous atrophy of the cranial small intestine (*Trichostrongylus*); subserosal nodules (0.5–1 cm diameter with a caseous or mineralized center) in the large intestine, and occasionally observed in liver, lungs, mesentery and mesenteric lymph nodes, presence of nematodes in the intestine (Oesophagostomosis); proliferative enteritis of the caecum, colon and small intestine associated with the presence of small greyish-white mucosal multifocal lesions (nodules) (1–2 mm diameter) (*Eimeria*)	
**Liver Fluke**		**References**
Etiology	*Fasciola hepatica*, *Dicrocoelium dendriticum*	[75,76,77]
Clinical signs (Antemortem)	Anemia, hypoalbuminemia, emaciation, submandibular edema (bottle jaw) and ascites, progressive loss of condition, and chronic emaciation.	
Gross lesions (Post-mortem)	Chronic cholangitis, bile duct obstruction owing to the presence of mature flukes, the liver visceral surface presents white, firm, branching cords resulting from periductular fibrosis and eventual mineralization, periportal and mesenteric lymph nodes markedly enlarged exhibiting brownish color, possible proliferative peritonitis, the appearance of fibrous tags with adhesions or diffuse fibrous thickening.	

**Table 5 animals-11-03028-t005:** Classification, disposal, and use of condemned wasting carcasses affected organ or part of the carcass (animal by-products), according to Regulation (EC) No 1069/2009 [27].

By-Products Categorization	By-Product	Disposal and Use of Animal By-Products
C1	- Carcasses, organs including hides and skins—wasting condition due suspected or confirmed scrapie	- Incineration or co-incineration
C2	- Carcasses and organs—wasting condition not related to TSE diseases - Condemned organs or part of the carcass – presenting gross lesions.	- As C1, or: - Disposed of in an authorized landfill, following processing - Manufacturing of organic fertilizers or soil amendments following processing - Composted or transformed into biogas.
C3	- Carcasses and organs—cachexia senile - Wasting carcass due to primary nutritional deficiencies.	- As C2, or: - Manufacturing of pet food.

## Data Availability

Not applicable.

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
