# Peer review of "Scrapie at Abattoir: Monitoring, Control, and Differential Diagnosis of Wasting Conditions during Meat Inspection"

_animals, 2021, doi:10.3390/ani11113028_

Round 1
Reviewer 1 Report
The Authors described in a review Scrapie at abattoir: Monitoring, control and differential diagno-2 sis of wasting diseases during meat inspection. In my opinion, the work is written in the correct language and understandable to the reader. A small spell check is required.
My comments are rather minor.
Line 21- Change to „It is common in slaughterhouse to see small ruminants showing signs of poor condition with .....”
Line 23- Change to „..of the carcass during post-mortem inspection, in order to prevent unfit meat entering human food chain”
Line 28 – change „Scrapie suspicion at” to „Scrapie suspicion in..”
Line 30: Change to „ post-mortem decision as well as animal by-product further processing”
Line 30: Change „viscera” to „offal”
Line 41 : Change „forms” to „types”
Line 50 : „Change to „ Ante-mortem and post-mortem inspection of animals is important ...”
Line 51 : Change „major” to „vital/important”
Line 59: Change „procedurÄ™” to „inspection”
Line 161 : change „perform” to „ developed and implemented”
Line 167: Add „human consumption”
Line 168: „Change to This program is under constant review”
Line 171” Change to „..for living livestock”
Line 394 This regulation was replaced by new regulation (REGULATION (EC) No 1069/2009 OF THE EUROPEAN PARLIAMENT AND OF THE COUNCIL of 21 October 2009 )
Author Response
The authors thank the reviewer for his/her suggestions and comments. This reviewer only had a few suggestions for improving and/or correcting, in a few places, what was presented. As such, all this reviewer’s suggestions were included in the revision of the manuscript, as most were relatively minor.
Regulation (EC) No 1774/2002 [88] was updated, as advised, and replaced by Regulation (EC) No 1069/2009 of the European Parliament and of the Council.
Reviewer 2 Report
- Introduction
Please include a sub-section to briefly mention the other wasting diseases of small ruminants.
Please described clearly the objectives of the paper.
A new section is necessary to present the methodology of detecting and selecting the papers that were reviewed.
- Section 2
Figure 1. Please use colours to enhance this figure.
There is no sub-section 2.2. ?
- Section 3
Figure justifies the necessity for a sub-section with wasting diseases as described above.
Table 3. Lungworms and gastrointestinal parasites are not causing wasting disease. They can cause poor condition, but NOT wasting disease. The authors have made a scientific error in there.
Table 4. These should be inserted in Table 2.
Nutritional disorders: again nutritional problems do not cause wasting disease. They cause poor body condition, yes – wasting disease, definitely no. This is again a scientific error.
- and 3.6. should be included into a separate section (no. 4)
General.
The manuscript needs further extensive work before resubmission. As it is now, it is now publishable.
The manuscript also misses many important references in the topic. These should be traced and reviewed.
Finally, 22 authors in a review paper? This is a bit confusing.
Author Response
The authors understand and appreciate the helpful comments and suggestions by the reviewer. The reviewer voiced a number of doubts concerning some aspects of the manuscript. We address these doubts, many of which can be explained by contextualizing some of the information presented to our situation and location.
Contextualization of the manuscript:
The authors consider it important to reveal the causes for the high number of small ruminants condemned at Portuguese slaughterhouses (official records) based on observation of a “wasting” condition (poor condition). The manuscript presents the need for appropriate diagnosis of this “wasting” condition so as to pinpoint cases in which scrapie is the cause.
The methodology used for determining diseases/conditions associated with wasting conditions, outlined in the manuscript, was based on reports emerging from several meetings with wide stakeholder participation. This group of stakeholders was multidisciplinary including academics, meat inspection researchers, official meat inspectors, veterinarians specialized in small ruminant diseases, pathologists, veterinary public health specialists and geneticists. This selection was, thus, based on the experience of these stakeholders, in addition to data from reports of official Portuguese organizations.
The establishment of this working group, included as authors of this manuscript, has been very fruitful, resulting in several insightful studies regarding anatomopathological and microbiological findings related to wasting and to other causes of condemned carcasses at slaughterhouses.
The authors’ particular interest in scrapie (classical and atypical) lies in the fact that these prion diseases have a particular impact on animal health. This impact justifies the importance of the differential diagnosis of these diseases in slaughterhouses especially relevant.
Methodology in the selection of articles used in this manuscript
For preparing this manuscript, we considered participation of a large number of selected stakeholders. In so doing, the text of the manuscript includes contributions for each disease/condition/disorder by one or more authors. Each of these authors took on the responsibility of providing cited references that were most relevant, important and updated, taking into consideration the scope of the article.
From this standpoint, this article is not intended to be a meta-analysis or a systematic review. So, at its inception, the authors did not consider it essential to describe the methodology used to select the papers that were reviewed, one of the comments of the reviewer.
However, the authors have now included in the manuscript the methodology regarding the selection of the considered diseases/conditions. Thus, the following paragraph was included in line 239 to line 244 of the revised manuscript:
“The selection of diseases/conditions associated with wasting conditions was established through thorough discussions with a group of multidisciplinary stakeholders (academics, meat inspection researchers, official meat inspectors, practitioners specialized in small ruminant diseases, pathologists, veterinary public health specialists and geneticists), based on the totality of condemnation data at abattoir and respective anatomopathological and microbiological findings, in Portugal.”
Wasting definition:
The authors agree with the comment of the reviewer concerning “wasting disease” v “condition”. We believe that this discordance was due to a misleading title. In fact, throughout the manuscript, the most used expression is “wasting condition” rather than “wasting disease”. Therefore, the authors have now changed the Manuscript Title from:
“Scrapie at abattoir: Monitoring, control and differential diagnosis of wasting diseases during meat inspection”.
To:
“Scrapie at abattoir: Monitoring, control and differential diagnosis of wasting conditions during meat inspection”.
-Also, between lines 74 and 98 of the manuscript, by mistake, parasitic diseases, tumors and nutritional disorders were treated as wasting diseases.
The authors have now amended this paragraph (underlined portions of the below paragraph), by distinguishing true chronic wasting diseases (Maedi-Visna, paratuberculosis, caseous lymphadenitis, tuberculosis) from those situations where, by definition, that are not wasting diseases, per se, but may in severe cases be accompanied by a wasting condition of the animal and its carcass:
“The wasting condition can be commonly observed in small ruminants during meat inspection. The wasting condition can result from poor condition, emaciation, or cachexia. Although emaciation and cachexia can result from similar etiologies, they are pathological conditions resulting in carcass condemnation during meat inspection.
In animals designated as thin, there is a marked reduction of fat, but, otherwise, the animal has a normal appearance and texture [2]. Emaciation, compared to thinness (the two should not be confused) is a pathological condition involving the atrophy of muscular tissue in addition to a reduction in the amount of fat tissue, which acquires an abnormal oedematous and jelly-like consistency with an off-color, yellowish appearance (serous fatty atrophy) [2–4]. Cachexia is the most severe case of extreme emaciation. With this condition there is also fatty tissue atrophy but additional atrophy of muscles and liver and shrunken lymph nodes [2,4]. In the most severe cases of cachexia, there is serous atrophy of bone-marrow fat resulting in it appearing as a red, gelatinous mass [2,3].
Several conditions and diseases can be related to emaciation, including scrapie (classical and atypical). (…) Generally, this condition is linked with chronic wasting diseases such as Maedi-Visna, paratuberculosis, caseous lymphadenitis, tuberculosis and severe cases of parasitic diseases, tumor diseases, toxic or metabolic conditions or inadequate nutrition due to poor teeth or poor alimentary diet.”
Responses to “Comments and Suggestions for Authors
”:
Section 1- Introduction
- Please include a sub-section to briefly mention the other wasting diseases of small ruminants.
Authors answer:
As it was mentioned above, the methodology used to the selection of diseases/conditions associated with wasting conditions considered in this manuscript, was based on information that emerged from several meetings in which different stakeholders associated with the subject-matter participated. This multidisciplinary stakeholders group consisted of academics, meat inspection researchers, official meat inspectors, veterinarians specialized in small ruminant diseases, pathologists, veterinary public health specialists and geneticists. This selection of the disease/conditions in the manuscript was made based on the experience of these stakeholders, as well as data from reports of official Portuguese organizations.
- Please describe clearly the objectives of the paper
Authors answer:
We consider that the below paragraph (adapted from Lines: 98-105) better reflects the objectives of the manuscript
“This manuscript reviews clinical signs and indicator lesions, to differentiate causes of wasting conditions in carcasses of small ruminants at the slaughterhouse. The differential diagnosis in the slaughterhouse of gross lesions associated with wasting condition is fundamental to food safety, including determination of treatment and destination of animal by-products. The main focus emphasizes differential diagnosis of putative scrapie relative to other wasting conditions/diseases. The review also covers recommended certified laboratory tests and measures at the abattoir for scrapie control.”
- A new section is necessary to present the methodology of detecting and selecting the papers that were reviewed.
Authors answer:
As stated above, the chosen methodology for preparing this manuscript included participation of a large number of stakeholders, whose area of interest and expertise are in the subject of the manuscript. Therefore, text for each disease/condition/disorder was provided by one or more authors, who provided a bibliography they chose as being the most relevant, important and updated, within the subject area of their contribution.
Hence, this article is not intended to be a meta-analysis or a systematic review. Thus, the authors, at the outset, did not consider it essential to describe the methodology used to select the papers that were reviewed.
We have now included, in the manuscript, the methodology for selecting the diseases/conditions considered. Thus, the following paragraph was included in line 239 to line 244 of the revised manuscript:
“The selection of diseases/conditions associated with wasting conditions was established through thorough discussions with a group of multidisciplinary stakeholders (academics, meat inspection researchers, official meat inspectors, practitioners specialized in small ruminant diseases, pathologists, veterinary public health specialists and geneticists), based on the totality of condemnation data at abattoir and respective anatomopathological and microbiological findings, in Portugal.”
Section 2
- Figure 1. Please use colours to enhance this figure.
Authors answer:
Colours have been added to figure 1. The authors used colours in figure 3, too, with the aim of rendering the figure schemes to be more coordinated.
- There is no subsection 2.2. ?
Authors answer: We deleted subsection 2.1.
Section 3
- Figure justifies the necessity for a sub-section with wasting diseases as described above.
Authors answer:
Lines: (231-234):
“A graphical categorization of the main wasting diseases/conditions, other than scrapie, seen during meat inspection is presented in Figure 2. Clinical signs (ante-mortem) and gross pathological findings (post-mortem) of these non-scrapie wasting conditions are described further, in the following sections (3.1 to 3.4).”
- Table 4. These should be inserted in Table 2.
Authors answer:
The authors appreciate the comment offered by the reviewer. However, the tumors described in Table 4 are of viral origin. The authors would like to present the “Tumors” separately since malignant tumors in sheep and goats are generally associated with declining condition, and in extreme cases with cachexia.
For this reason, we would like to maintain Table 4, where the most frequent tumors found in Item 3.3 are described.
- Lungworms and gastrointestinal parasites are not causing wasting disease. They can cause a poor condition, but Not wasting disease. The authors have made a scientific error in there.
- Nutritional disorders: again nutritional problems do not cause wasting diseases. They cause poor body condition, yes – wasting diseases, definitively no. This is again a scientific error.
Authors answer:
The authors thank the reviewer for pointing this discrepancy out; which resulted in our changing the title. Throughout the manuscript, we mainly used the expression “wasting condition” rather than “wasting disease”. Therefore, the authors have now changed the Manuscript Title from:
“Scrapie at abattoir: Monitoring, control and differential diagnosis of wasting diseases during meat inspection”.
To:
“Scrapie at abattoir: Monitoring, control and differential diagnosis of wasting conditions during meat inspection”.
-Also, between line 74 and 98 of the manuscript, by mistake, parasitic diseases, tumors and nutritional disorders were treated as wasting diseases.
The authors have now amended this paragraph (underlined portions of the below paragraph), by distinguishing true chronic wasting diseases (Maedi-Visna, paratuberculosis, caseous lymphadenitis, tuberculosis) from those situations where, by definition, that are not wasting diseases, per se, but may in severe cases be accompanied by a wasting condition of the animal and its carcass:
“The wasting condition can be commonly observed in small ruminants during meat inspection. The wasting condition can result from poor condition, emaciation, or cachexia. Although emaciation and cachexia can result from similar etiologies, they are pathological conditions resulting in carcass condemnation during meat inspection.
In animals designated as thin, there is a marked reduction of fat, but, otherwise, the animal has a normal appearance and texture [2]. Emaciation, compared to thinness (the two should not be confused) is a pathological condition involving the atrophy of muscular tissue in addition to a reduction in the amount of fat tissue, which acquires an abnormal oedematous and jelly-like consistency with an off-color, yellowish appearance (serous fatty atrophy) [2–4]. Cachexia is the most severe case of extreme emaciation. With this condition there is also fatty tissue atrophy but additional atrophy of muscles and liver and shrunken lymph nodes [2,4]. In the most severe cases of cachexia, there is serous atrophy of bone-marrow fat resulting in it appearing as a red, gelatinous mass [2,3].
Several conditions and diseases can be related to emaciation, including scrapie (classical and atypical). (…) Generally, this condition is linked with chronic wasting diseases such as Maedi-Visna, paratuberculosis, caseous lymphadenitis, tuberculosis and severe cases of parasitic diseases, tumor diseases, toxic or metabolic conditions or inadequate nutrition due to poor teeth or poor alimentary diet.”
- And 3.6. should be included into a separate section (no.4)
Authors answer:
Point 3.6 was transformed into 4 and Conclusions into 5.
General
- The manuscript also misses many important references in the topic.
- Finally, 22 authors in a review paper? This is a bit confusing.
Authors answer:
As explained above: The chosen methodology for preparing this manuscript considered the participation of a wide range of stakeholders having expertise and experience on the subject of the manuscript. Text for each disease/condition/disorder was provided by one or more authors with respect to the subject are of their expertise. Each author provided an updated bibliography most relevant to the scope of the article.
At its inception, this article was not intended to be a meta-analysis or a systematic review. As such, the authors did not consider it essential to describe the methodology used for selecting the papers that were reviewed.
To the Editor:
The authors wish to acknowledge the thorough review and valid contribution of both reviewers. Their reviews contributed to our improving the clarity and focus of the manuscript.
Regarding reviewer 2, we made sincere attempts to meet the expressed comments and requests. However, in a few cases, there were constraints in meeting the reviewer’s suggestion for changing the original manuscript, as explained and demonstrated in our answers, stated above.
We trust we have revised the manuscript sufficiently to adequately incorporate the reviewers’ suggestions. We thank you for the opportunity to revise our manuscript to the point where you can consider it for publication.
Reviewer 3 Report
No comments
Author Response
Dear Editor
The authors thank to reviewers and editors for their suggestions and comments.
We have considered all suggestions.
We are sending the revised document with the track changes, and clear.
We hope that our manuscript “Scrapie at abattoir: Monitoring, control and differential diagnosis of wasting conditions during meat inspection” will be considered for publication in the special issue “Diseases of Sheep”.
Thank you
We look forward to hearing from you.
Yours sincerely.
Vila Real, 15th October 2021
On behalf of all authors,
Maria dos Anjos Pires
Associated Professor with Habiliti,
Veterinary Sciences Department
ECAV, CECAV, Lab4Animals, UTAD
Portugal
Round 2
Reviewer 2 Report
The authors have corrected the manuscript, which has been significantly improved.
Nevertheless, I still have two issues.
First. The change regarding wasting problem is not acceptable, simply because the approach is not correct. If the authors wish to maintain this scientific error in their manuscript, it will be their choice, it is their names that they put at stake.
Second. The main text is 440 lines and includes 86 references. With 22 authors, it means that on average each author has contributed 20 lines. of text and 4 references. In reality, there are authors who contributed 1 or 2 references and 5-7 lines of text and these obviously do not justify authorship. The situation also raises questions regarding the competence of the main authors: could they not find the references themselves? Could they not comprehend and discuss the references themselves? Then, they should not have written the review in the first place.... After publication, many readers will ask the same questions and the authors will not able to provide answers, hence the readers will believe that there were gift-authorships in this manuscript. Is that what the authors want? To be characterised that they distribute and accept gift-authorships? Again, this should be the decision of the authors.
Opinion: The academic editor can decide upon these issues. Minor revision.
Author Response
The Authors acknowledge the reviewer`s comment.
First. The approach used in the manuscripts inception was not that different from the one that can be found in the webpage:
https://www.mdpi.com/journal/animals/special_issues/Wasting_Diseases_Affecting_Sheep
regarding: “A special issue of Animals (ISSN 2076-2615).”, a few months ago.
The text includes the following paragraph:
“Although chronic wasting diseases in sheep may be associated with management problems or nutritional deficiencies, there are several infectious diseases that occur with this symptomatology, such as Maedi-visna, paratuberculosis, caseous lymphadenitis, scrapie, or respiratory tumors from ovine pulmonary adenocarcinoma or enzootic nasal adenocarcinoma, most of them widespread worldwide. Further, other toxic or nutritional diseases, mineral deficiencies, as well as oral disorders, can lead to chronic weight loss. Of course, the importance of parasitic diseases in the development of this syndrome cannot be ignored.”
Nevertheless, trying to approach the reviewer`s suggestion, on section “3.2. Parasitic diseases”, on line 270 of the manuscript – version “animals-1363624 for second review”, it was deleted the expression “wasting and”.
Second. The Authors acknowledge the reviewer`s comment. However, as explained before, the chosen methodology for preparing this manuscript considered the participation of a wide range of stakeholders having expertise and experience on the subject of the manuscript. Text for each disease/condition/disorder was provided by one or more authors with respect to the subject are of their expertise. Each author provided an updated bibliography most relevant to the scope of the article.